# Implementation of an Effective Decentralised Programme for Detection, Treatment and Prevention of Tuberculosis in Children

**DOI:** 10.3390/tropicalmed6030131

**Published:** 2021-07-14

**Authors:** John Paul Dongo, Stephen M. Graham, Joseph Nsonga, Fred Wabwire-Mangen, Elizabeth Maleche-Obimbo, Ezekiel Mupere, Rodrigo Nyinoburyo, Jane Nakawesi, Gerald Sentongo, Pauline Amuge, Anne Detjen, Frank Mugabe, Stavia Turyahabwe, Moorine P. Sekadde, Stella Zawedde-Muyanja

**Affiliations:** 1International Union against Tuberculosis and Lung Disease, Kampala P.O. Box 16094, Uganda; jpdongo@theunion.org (J.P.D.); jnsonga@theunion.org (J.N.); 2Centre for International Child Health, Department of Paediatrics and Murdoch Children’s Research Institute, University of Melbourne, Royal Children’s Hospital, Melbourne 3052, Australia; 3School of Public Health, Makerere University College of Health Sciences, Kampala P.O. Box 16094, Uganda; fwabwire@musph.ac.ug; 4Department of Paediatrics and Child Health, School of Medicine, University of Nairobi, Nirobi P.O. Box 30197-00100, Kenya; eobimbo@yahoo.com; 5Department of Paediatrics and Child Health, School of Medicine, Makerere University College of Health Sciences, Kampala P.O. Box 7072, Uganda; mupez@yahoo.com; 6Mildmay Uganda, Kampala P.O. Box 24985, Uganda; rnyinoburyo@rhites-ec.urc-chs.com (R.N.); jane.nakaweesi@mildmay.or.ug (J.N.); 7Baylor College of Medicine Children’s Foundation-Uganda, Kampala P.O. Box 72052, Uganda; sentongog@gmail.com (G.S.); pamuge@baylor-uganda.org (P.A.); 8Unicef, New York, NY 10017, USA; adetjen@unicef.org; 9National Tuberculosis and Leprosy Program, Kampala P.O. Box 16069, Uganda; rfmugabe@gmail.com (F.M.); turyahabwestavia@gmail.com (S.T.); moorine.sekadde@gmail.com (M.P.S.); 10The Infectious Diseases Institute, College of Health Sciences, Makerere University, Kampala P.O. Box 22418, Uganda; szawedde@idi.co.ug

**Keywords:** tuberculosis, child, case detection, contact management, implementation

## Abstract

Childhood tuberculosis (TB) is consistently under-detected in most high-burden countries, including Uganda, especially in young children at high risk for severe disease and mortality. TB preventive treatment (TPT) for high-risk child contacts is also poorly implemented. The centralised concentration of services for child TB at the referral level is a major challenge in the prevention, detection and treatment of TB in children. In 2015, the DETECT Child TB Project was implemented in two districts of Uganda and involved decentralisation of healthcare services for child TB from tertiary to primary healthcare facilities, along with establishing linkages to support community-based household contact screening and management. The intervention resulted in improved case finding of child and adult TB cases, improved treatment outcomes for child TB and high uptake and completion of TPT for eligible child contacts. A detailed description of the development and implementation of this project is provided, along with findings from an external evaluation. The ongoing mentorship and practical support for health workers to deliver optimal services in this context were critical to complement the use of training and training tools. A summary of the project’s outcomes is provided along with the key challenges identified and the lessons learnt.

## 1. Background

It is estimated that *Mycobacterium*
*tuberculosis* causes disease in around 1 million children (<15 years) globally each year and over 200,000 deaths [1,2]. It is widely recognised that major contributors to the global burden of child tuberculosis (TB) are under-detection and treatment, especially in young children (<5 years), along with the low coverage of effective TB preventive treatment (TPT) [3,4]. In most countries, services for child TB are centralised with expertise concentrated at the referral level and not well integrated with other child health services. However, most opportunities for early detection, treatment and prevention of TB in children occur at the household or peripheral health facility level [5,6].

Uganda reports a high incidence of TB and TB/HIV—200 and 81 per 100,000 population, respectively, in 2018 [1]. Although they comprise around 50% of the country’s population [7], children represent less than 15% of all TB cases reported to the Ugandan National Tuberculosis and Leprosy Programme (NTLP), which is lower than that expected for a country with high TB incidence with such age demographics [1]. The major barriers to detection are well recognised and include diagnostic challenges, not only for bacteriological confirmation but also for clinical diagnosis by healthcare workers, and the poor coverage of active case finding in high-risk children such as those who are household contacts of patients with active TB. A previous study in Uganda reported a high yield of active case finding in child contacts—around 10% of 761 child contacts had active TB, most (71%) with bacteriologically confirmed TB [8]. Disease prevalence was highest in young child (<5 years) contacts, equating to 16,400 per 100,000 young child contacts.

In 2014, the DEcentralise Tuberculosis services and Engage Communities to Transform lives of Children with TB (DETECT Child TB) project was initiated in Uganda [9]. This project developed a globally applicable model for the integration of child TB services into child healthcare services and TB programme activities. The project was piloted in two distinct districts of Uganda (Figure 1) and utilised materials and guidelines developed by the International Union against Tuberculosis and Lung Disease (The Union) and the World Health Organisation (WHO) [10,11,12,13]. Following ethical approval, the DETECT Child TB project was implemented in 2015, and a detailed external evaluation was undertaken at the end of 2016. A summary of the main project outcomes is listed in Table 1, of which the quantitative analysis findings are published [14].

To our knowledge, this is the first such model to be implemented and evaluated. There is potential applicability of the decentralised delivery model to other resource-limited TB-endemic settings, which is further emphasised in the context of the current coronavirus pandemic. We, therefore, aim to provide details of steps for implementation, the main project outcomes including external evaluation findings, challenges encountered and addressed and lessons learnt.

## 2. Project Implementation

### 2.1. Project Design

The DETECT Child TB project was an integrated model of care that deliberately decentralised child TB care services from tertiary referral hospitals. The project focused on primary and secondary care facilities where the majority of children with TB initially present, aiming to improve facility-based healthcare worker capacity to undertake clinical assessment and diagnostic work-up of children with presumptive TB and to initiate treatment in children with a diagnosis of TB. The project created linkages between the health facility and community-based services to allow for continuity of care.

### 2.2. Project Setting

The project was implemented in two districts in Uganda with locations shown in Figure 1. Kabarole District is a rural district in western Uganda with a population of around 500,000 people and Wakiso District is peri-urban with a population of nearly two million people. These districts were selected because they had existing health system support structures for the delivery of HIV care services for children and adolescents, and both had poor indicators for child TB with low case detection, treatment success rates and TPT coverage [14]. All health facilities offering general TB diagnostic and treatment services in the two districts were included.

### 2.3. Programmatic Engagement and Collaborative Partnership

Prior to the start of project implementation, the Uganda NTLP had identified the scale-up of child TB services as an important focus in the National Strategic Plan, 2015 to 2020. To realise this, the NTLP appointed a national child TB focal person (M.P.S.) and developed national TB management guidelines and child TB training materials. The Baylor College of Medicine Children’s Foundation (Baylor-Uganda) and Mildmay Uganda were implementing comprehensive HIV treatment projects that had large components of paediatric care in the Kabarole and Wakiso districts. To implement the DETECT Child TB project, we, therefore, formed a consortium of four partners: The Union-Uganda, the NTLP, Baylor-Uganda and Mildmay Uganda. The Union-Uganda obtained funding support and ethical approval, provided overall technical leadership and financial management and led the capacity building, health systems strengthening and project evaluation components. The NTLP provided policy frameworks for project implementation, e.g., consistency with national child TB management guidelines, for monitoring project performance and for mobilising resources for the scale-up of successful interventions. Baylor-Uganda and Mildmay Uganda worked together with the respective district health offices to implement project activities at the health facilities. All partners contributed to training, mentorship and ongoing technical support.

### 2.4. Project Activities

We built on the existing structures at the NTLP to conduct this pilot project with components of the intervention package including the following:collecting baseline data for the project districts;conducting stakeholder engagement meetings at the national and district levels;training of facility and community teams;supporting and strengthening the health systems delivery; andreviewing and optimising performance management at all levels.

#### 2.4.1. Baseline Survey

We conducted a baseline survey in both districts before project inception in April 2015. Data were collected on current practices regarding prevention, diagnosis and treatment of TB in children and on existing community structures for TB care. Results from the survey showed that children (<15 years) contributed to less than 10% of the total TB caseload notified from July 2013 to June 2014 in Kabarole and Wakiso districts. Almost all child TB cases were being diagnosed at the referral (tertiary) hospital, and no community structures existed for the prevention or management of child TB.

#### 2.4.2. Stakeholder Engagement

A national meeting was held in Kampala to introduce the project to relevant stakeholders including representatives from the Ministry of Health, district local governments (both the political leadership and the district health teams) and development partners, including the Centres for Disease Control and Prevention (CDC), the United States Agency for International Development (USAID) and non-governmental organisations implementing healthcare interventions in Uganda. At this meeting, results from the baseline survey were disseminated. Similar engagement meetings were held at each district. The district leadership agreed to implement project activities under the coordination of the district health officer, and the roles of the district teams in supporting the project were discussed and agreed.

#### 2.4.3. Training of Healthcare Workers and Community Teams

(a)Training materials developed

Training tools such as teaching presentations, flipcharts and evaluation tests were developed for both health facility and community healthcare workers. The materials were consistent with the Uganda national childhood TB guidelines and adapted to the local context from The Union’s Desk-guide and online child TB training course [10,13].

(b)Training provided

Fifteen district-level trainers were trained on child TB management from each district. These district trainers were responsible for training healthcare workers at the primary health facilities with oversight supervision from The Union and its partners. District trainers, therefore, required a clear understanding of the district health system, and so were mostly selected from established members of the District Health Team. The two-day didactic training focused on the diagnosis and treatment of TB in children and on contact screening, prevention and management. The didactic sessions were reinforced with numerous case studies adapted from the Union’s online childhood TB course. A pre- and post-training evaluation of knowledge was conducted. Training for diagnosis also included practical training on sputum induction and gastric lavage at 14 hospitals and level IV health facilities in both districts. This skills-based training was conducted by two nurses from the National Referral Hospital, and following training, equipment to support the procedures was supplied to the hospitals.

Community health workers were identified and linked to a health facility (two each per facility). They were trained to perform screening of household contacts of people with bacteriologically confirmed TB; identify and refer symptomatic contacts of any age for further evaluation; refer asymptomatic young child (<5 years) contacts to health facilities for initiation of TB preventive therapy (TPT), specifically six months of daily isoniazid (6H)—as per NTLP guidelines, and to provide families with health education and TPT adherence support (Figure 2).

(c)Post-training follow-up support

Following the didactic training, the district trainers provided ongoing support and mentorship to the healthcare workers. The continued mentorship aimed to reinforce the skills gained during the training and to help address health systems challenges that hindered service delivery. To enhance acquired competencies and confidence, on-site mentorship sessions were conducted monthly for the first three months after the didactic training and then quarterly, while ensuring minimal interruption of healthcare delivery at the facilities. During on-site mentorship, healthcare workers were introduced to The Union’s online training course on child TB. We supported healthcare workers to enrol for and complete the six-module child TB course during weekly continuous medical education sessions, which were facilitated by the project staff.

#### 2.4.4. Health Systems Strengthening Support

At the start of the project, some facilities in both districts had faulty laboratory equipment, and so TB diagnostic services could not be provided at the health facility. To restore and maintain these services, we repaired faulty microscopes, retrained laboratory personnel on acid-fast bacilli microscopy and then enlisted quarterly external quality assurance for sputum microscopy through NTLP. At the time of the implementation project, WHO-approved rapid diagnostics such as Xpert were not available except at the referral (level V) facility level. Stock-outs of medicines and laboratory supplies were also identified as common challenges for service delivery in some facilities, especially in Wakiso District. We provided supply chain management training to facility-based healthcare and procured additional supplies of medicines, such as isoniazid-only preparations for use as TPT to buffer government stocks, which had intermittent stock-outs throughout the project.

#### 2.4.5. Optimising Performance Development

We conducted quarterly review meetings at each district to periodically evaluate the performance of facility- and community-level interventions. Health facility staff used a *continuous quality improvement approach* to identify challenges and give feedback on the DETECT Child TB interventions to both district and the project teams. The meetings provided a platform for developing solutions to identified challenges for project improvement and provided a suitable opportunity to share best practices among various stakeholders at the district level. Priorities for the coming period were discussed and agreed on. Health facility staff also conducted quarterly community linkage meetings between community health workers and the health facility team. At these meetings, teams evaluated the community workers’ performance and addressed community-level challenges.

### 2.5. Project Data Management

Each quarter, the district TB officers manually extracted data from the health facility TB registers—presumptive TB register, TB treatment register and IPT register. Data were entered into Microsoft Excel. Project officers then validated and de-identified the data. Key indicators collected included the number of index TB patients, number of child TB cases and child TB treatment outcomes. We collected additional data on contact screening and management from the contact screening registers, which were introduced by the project in the two districts. The key cascade of care indicators collected from these tools included the following: number of bacteriologically confirmed index TB patients for whom contact screening was carried out; number of child contacts screened, number of child contacts with a positive screen for TB signs and symptoms, number of child contacts asymptomatic and eligible for TPT, number of eligible child contacts initiated on TPT and the number completing TPT. Project outcomes within the pilot districts were evaluated using before-and-after analysis, and comparisons of child TB data and indicators were made with two similar non-intervention districts—Hoima and Mukono Districts.

### 2.6. Project Evaluations

A mid-term evaluation was undertaken eight months after initiation of the project by one external child TB expert who had an excellent working knowledge of the Uganda NTLP. The evaluator carried out monitoring visits in both districts, met with the project and district local leaders and provided a detailed report with feedback to project leaders in Kampala.

An extensive evaluation of the project was undertaken at the end of 2016 by external national (F.W.-M. and E.M.) and international (E.M.-O.) consultants with expertise in epidemiology, child TB and public health as well as an excellent working knowledge of Uganda and the region. A mixed-methods approach was used that consisted of a desk review of pertinent literature and reports, secondary analysis of NTLP data on child TB outcomes and indicators and field work to assess project implementation using qualitative and quantitative methods. A survey was conducted at health facilities in the two implementing districts (Wakiso and Kabarole) and the two comparative non-implementing districts (Hoima and Mukono) using a tool adapted from the baseline survey, which allowed a before-and-after comparison of key childhood TB outcomes and indicators. The evaluation gathered information through reports and data review, interviews with project staff and local implementing partners, as well as focus group discussions and interviews with consumers such as district personnel, local service providers and community health workers. The consultants developed data collection and evaluation tools for ensuring consistency of information for different target groups and field visits. In addition to a face-to-face debrief and feedback presentation of findings, a final report was submitted [15].

## 3. Main Project Outcomes

The main outcomes from data analysis and external evaluation of the DETECT Child TB project are listed in Table 1. A total of 30 district-level trainers were trained (five-day course) who then provided training on child TB to 193 frontline healthcare workers in the primary care facilities through a three-day course. In addition, 178 community healthcare workers (known as Village Health Teams in Uganda) were also trained to implement the screening and management of household contacts of TB cases through a two-day training course. There was evidence of successful decentralisation of child TB services with improved TB diagnostic capacity in 76 primary healthcare facilities. There were significant improvements in the detection of child TB, which healthcare workers largely attributed to their improved ability and confidence in making a clinical diagnosis of TB, and TB treatment outcomes for children and the successful introduction of household TB contact screening by community health workers with high uptake and completion of TPT for eligible child contacts [14]. It was also noted that the project led to strengthened services and TB case notifications for all ages in the intervention districts.

The feedback from respondents to focus group discussions and interviews was very positive [15]. When asked to gauge the performance of their district in line with Ugandan child TB guidelines, the majority of key informants scored their performance as very good and noted that the DETECT Child TB project had brought stakeholders together, educated on best practices and had strengthened health service capacity and delivery. Healthcare workers gained confidence to implement child TB activities and understood the child TB programme performance indicators and reporting. Availability of trained district health teams, health facility staff and community health workers enabled the integration of activities into routine TB programmes at the sub-national level even after the child TB project ended. However, it was noted that continued support to the districts such as through ongoing on-site mentorship would be required to sustain the success of the intervention. Key informants felt that it would be difficult for their district to sustain transport and phone card costs, especially for the village health teams.

The key challenges and lessons learnt as identified by the external evaluation are listed in Table 2. Key recommendations listed in the report [15] were that, in collaboration with NTLP, there should be:strengthening of child TB health service delivery capacity through:
○building confidence of healthcare workers to diagnose, manage and report TB, including TB in children, with continuous mentorship and supervision being vital for building capacity and capability;○establishing and maintaining laboratory capacity for TB diagnosis in peripheral health facilities;○consistent availability of anti-TB medicines, including formulations suitable for TPT for children; and○training and linking community healthcare workers to primary care facilities and supporting them to integrate treatment support and education for the index case with household contact screening and management for case detection and prevention.
Decentralisation of child TB services with strengthening of capacity for detection, treatment and prevention at peripheral health facilities through:
○training and support of healthcare workers, including use of the Union’s child TB tools such as the online child TB course, and in-service updates to improve and maintain clinical diagnosis;○improved access to diagnostics to aid clinical diagnosis of TB among children, e.g., chest radiographs for children and strengthening of healthcare worker capacity to interpret them; and○improved capacity for sample collection in children and laboratory diagnosis, including wider availability and use of GeneXpert.


## 4. Discussion

The main component of the DETECT Child TB project was to decentralise child TB services from being concentrated at tertiary or referral facilities. This process required upskilling of health workers based at the more peripheral levels to develop the capacity to confidently diagnose, treat and prevent TB in children. Prior to the project, health workers in the lower-level facilities did not have sufficient knowledge and skills, which led to either under-detection (and -treatment) or to referral of presumptive child TB cases to often distant, specialised paediatric services [14]. Strengthening capacity at the primary care level then facilitated the successful introduction of community-based household contact screening and management.

Establishing capacity for service delivery at the primary care and household level greatly reduces the current wide policy–practice gap by implementing the WHO’s comprehensive End TB Strategy and reducing losses along the cascades of care for detection, treatment and prevention [5]. It is well documented that current common practices or approaches, such as the referral for care of all children with presumptive TB to a more central facility or the passive, health-facility-based approach to child contact screening and management, are associated with poor uptake and outcomes as well as potentially increased costs to families and health services [16,17]. An important component of this project was the strengthening of the district health systems that provided an enabling environment for healthcare workers such as by ensuring that laboratory, information and commodity supply systems function optimally. This component has enabled the districts to sustain child TB interventions beyond the pilot project.

The ongoing support and mentorship provided by the project, along with regular review and quality improvement efforts were critical to capacity development and sustained, effective implementation. The competency-based training materials used in this project were developed into a comprehensive manual for the training of healthcare workers in child TB management. This manual was used by the NTLP to train healthcare workers countrywide. Following the nationwide training, the proportion of child TB cases among all cases notified countrywide increased from 7.4% to 12.5%. Whilst the majority of child TB cases are clinically diagnosed, the recent roll-out of Xpert in Uganda is likely to further increase the numbers and proportion of bacteriologically confirmed TB cases in children [6]. Xpert is recommended by the NTLP for child TB diagnosis in preference to smear microscopy, and access to the Xpert diagnostic tool has recently increased, including at peripheral levels of care.

The NTLP has used lessons and adapted best practices from the DETECT Child TB project to refine its child TB activities in the National Strategic Plan 2015–2020 and to inform the national model for household contact investigation developed in 2017. The contact-screening tools introduced by the DETECT Child TB project were adopted by the NTLP in 2016/2017 and rolled out countrywide. The results of this project were incorporated into the Uganda Global Fund grant application 2018/2020 for which the NTLP secured funding to roll out the DETECT Child TB model to ten districts with a high TB incidence and poor child TB indicators. Scale-up of this model through the Global Fund in the ten districts has increased child TB case notification by 27% [18]. Additional funding was secured under the 2021–2023 Global Fund grant to sustain implementation of the model in the initial ten districts and scale-up to forty additional districts.

The DETECT project model and findings have been shared and well received by other countries in the WHO AFRO region [19]. The project findings have informed a number of ongoing research activities within the region. The project’s best practices and training tools and methods are an integral part of the sub-Saharan Africa Region’s child and adolescent TB Centre of Excellence [20] recently established by The Union with support from the U.S. Centres for Disease Control. A separate project supported by The Union was recently conducted in four francophone African countries and reported successful implementation of a short TPT regimen for young child contacts with high rates of uptake and completion [20].

In conclusion, the decentralisation of child TB services with an emphasis on clinical diagnosis at lower-level health facilities was a viable strategy for increasing child TB case finding and improving treatment outcomes. Training, continuous mentorship and health system strengthening were vital in building and maintaining the capacity of healthcare workers to diagnose and manage childhood TB. The utilisation of community health workers was vital for improving contact screening and implementation of TPT.

## Figures and Tables

**Figure 1 tropicalmed-06-00131-f001:**
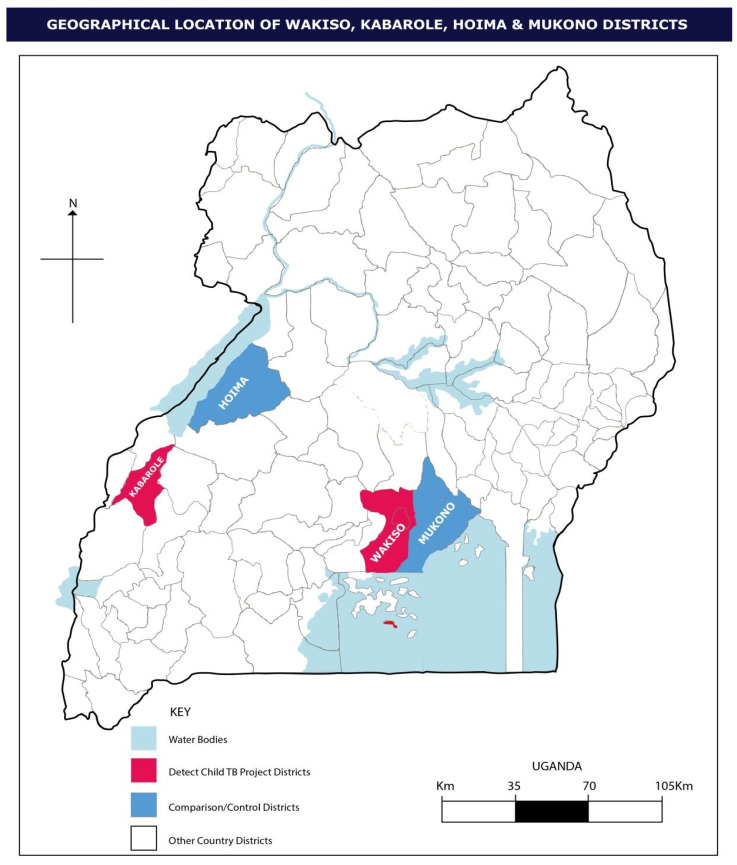
Map of Uganda showing the location of the DETECT Child TB participating districts and the control districts.

**Figure 2 tropicalmed-06-00131-f002:**
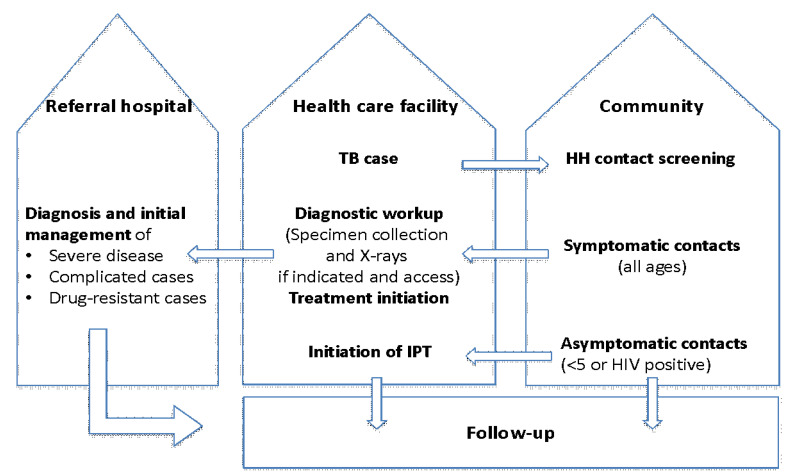
System of referral and patient management between the community and healthcare facilities. HH: household; IPT: isoniazid preventive treatment.

**Table 1 tropicalmed-06-00131-t001:** Summary of findings from evaluation of DETECT Child TB Project [14,15].

Project Outcome	Data Analysis #	External Evaluation Findings
**Decentralisation and Strengthening of Child TB Services**		
Proportion of all diagnosed child TB cases by health facility level *	Baseline: 96% at level V, 3% at level IV and 1% at level III By end implementation (Q4 2016): 50% at level V, 21% at level IV and 29% at level III	The DETECT model demonstrated that child TB services can be successfully decentralised with the greatest increase in detection occurring at the level III facility in both districts.
Health worker knowledge, Wakiso District	Average (range) test score Baseline: 40 (28–70)% Implementation: 75 (64–89)%	The ability and confidence of health workers in peripheral health facilities to diagnose TB in children was improved and the numbers of of unnecessary referrals were reduced.
Health worker knowledge, Kabarole District	Average (range) test score Baseline: 45 (32–64)% Implementation: 78 (68–96)%	
Number of functional TB basic management units for diagnosis and treatment by district	Baseline: 24 in Kabarole and 41 in Wakiso District By end implementation: 30 in Kabarole and 46 in Wakiso District	Repair of non-functional microscopes and re-training of laboratory personnel benefit TB services and detection for all ages. Improved laboratory and drug supplies with reduced stock-outs.
**Changes to TB Case Detection and Treatment Outcomes**		
Caseload of child TB and as a proportion of total TB notifications	139% increase in 0–14 years Baseline: 271, 8.8% Implementation: 647, 14.9%	The majority of respondents from the focus group discussions and in-depth interviews felt that the project had improved delivery of TB services, as it had accomplished the following: built capacity of health workers; increased detection of child TB cases; increased confidence in child TB management; strengthened facility and community household contact-tracing activities; built and increased community trust in healthcare workers through integration; contributed to a reduction in deaths in children. No respondents thought that the project was a burden to the community.
TB cases in young children, proportion of all child TB cases	Increase in <5 years ageBaseline: 99, 36.5% Implementation: 324, 50.1%	
Bacteriologically confirmed (BC) TB cases in children	61% increase in BC cases detected, but proportion with BC remained low in children. Baseline: 44 (9 young) children 0–14 years: 44/171 = 16% BC <5 years: 9/99 = 9% BC Implementation: 71 (16 young) children 0–14 years: 71/647 = 11% BC <5 years: 16/324 = 5% BC	
Cases of TB in older adolescents and adults	32% increase Baseline: 2805 Implementation: 3693	
Treatment success, cure or treatment complete	Significant improvement Baseline: 65% Implementation: 81%	
Died or treatment failure	Reduction in poor outcomes: Baseline: 15% Implementation: 4%	
**Household Child Contact Screening and Management ^**		
Training and support of community health workers to implement	178 (target was 168) community health workers received training	At least two per facility were trained over 2 days, provided with job aides and recording tools, linked with the facility-based TB focal person for ongoing mentorship and supervision.
Households screened	1617 households with 2270 child contacts	The project increased screening of households of smear-positive TB patients by 142%.
Child contacts with positive symptom screen	602 (27%) of 2270 child contacts were symptomatic	Screening symptoms used were cough, weight loss or poor weight gain, fever or lethargy/reduced playfulness.
Child contacts evaluated for TB disease	486 (81%) of 602 symptomatic child contacts	19% of symptomatic child contacts did not present to the health facility for further evaluation.
Child contacts diagnosed with TB	55 child TB cases detected	Lower numbers than expected for overall case detection—2.4% of all child contacts. Of symptomatic child contacts who presented to the facility for evaluation, 11% diagnosed with TB.
Child contacts eligible for TPT	910 young child contacts without active TB identified	Challenges with availability of isoniazid-alone preparation for TPT at beginning of project.
Eligible child contacts who initiated IPT	670 or 77% of 910 eligible	Remarkable improvements in IPT uptake noted in both districts over time but lower than the 90% target.
Child contacts who completed IPT	569 or 85% of 670 who commenced IPT	Although short of the 90% target, a high rate of completion.

TB: tuberculosis; TPT: tuberculosis preventive treatment; IPT: isoniazid preventive treatment. # Comparison was between two separate 18-month periods: baseline or pre-implementation was January 2014–June 2015; and implementation was July 2015–December 2016. * Level V: regional referral hospital for the district; level IV: secondary level health facility with laboratory and inpatient care; level III: primary level health facility with sputum smear microscopy available. ^ Routine household screening by community health workers was introduced by the project, and so there were no baseline programmatic data.

**Table 2 tropicalmed-06-00131-t002:** Challenges identified and lessons learnt from the DETECT Child TB Project.

**Key Challenges and Responses**
Feasibility for expansion: The human resources and investment required to establish and scale-up effective decentralisation of services to additional districts or to national coverage may be challenging in a resource-limited setting. A formal evaluation of the cost-effectiveness of the completed pilot project is underway. Sustainability: The time and effort required to provide training of trainers and then training for healthcare workers at primary care and community, followed-up by continuous quality improvement efforts through regular meetings and support may be difficult to sustain. The DETECT Child TB Project model is being applied in other districts in Uganda and has informed operational research and programmatic activities in other countries in the region. Diagnostic support: Although appropriate sampling techniques and laboratory services for bacteriological confirmation remain important, the main emphasis for improving diagnosis of child TB should focus on improving healthcare worker skills to make a clinical diagnosis of TB and increasing availability to diagnostic to aid clinical diagnosis, e.g., chest X-ray. Treatment: TB treatment and TPT dosage guidelines by weight bands must be available as well as appropriate preparations for young children, such as child-friendly fixed-dose combinations for treatment and single-drug formulations for TPT. The fixed-dose combinations for young children are now widely available, and the preparation that is used for continuation phase treatment (4RH) is also suitable, effective and safe as TPT (3RH) for young child contacts without active TB or HIV. Documentation of household contact tracing activities was manually done, which posed challenges in tracking child TB contacts referred from the households to the health system. However, this improved with continued mentoring and supervision of community healthcare workers. Challenges for household contact screening and management included the following: ○ Low case detection especially as symptomatic child contacts referred from households did not always present for further evaluation. ○ HIV testing of well contacts without symptoms was not performed, and so there were missed opportunities for people living with HIV to receive IPT. ○ Refusal by parents to give IPT, usually because they were not convinced that their well child should receive medicine daily for months.
**Lessons learnt**
Sustainability and scalability: District health teams and political leadership stated that the input and resources provided by implementation partners were pivotal to the success of the DETECT Child TB intervention and, thus a requirement for it to be sustained in the two pilot districts or to be successfully implemented in other districts. Health system strengthening: Training in a workshop setting followed by mentoring and supervision were crucial to effect lasting improvement in healthcare worker confidence and competence for child TB care. Wider capacity strengthening: An integrated approach can provide important benefits for all aspects of TB detection and care beyond the primary focus of the intervention, which in this project was the detection, treatment and prevention of TB in children. Training tools: The Union’s resources and online course on child TB were highly valued for initial group training and continuous in-service updates for healthcare workers. Decentralised detection of child TB: The diagnosis of TB in children, including clinical diagnosis in young children, can be achieved at the primary and secondary health facility level where most sick children with TB initially present. Decentralised treatment of child TB: TB in children can be successfully treated with a first-line treatment regimen at the primary and secondary health facility level. Community-based contact screening and management: Community healthcare workers can be successfully engaged to provide integrated care for household contacts of TB cases, including the detection and referral of symptomatic child contacts. Preventive treatment: Linkage of household contacts with primary care facilities through community healthcare workers can achieve high rates of uptake and completion of TPT. Coordination, communication and management through the consortium of partners was vital for the success of the project.

## Data Availability

All data from the research project are deidentified and securely stored at the Ugandan office of The Union in Kampala, Uganda. The data presented in this study are available on request from the first (J.P.D.) or corresponding author. The data are not publicly available due to national data policies and regulations that do not permit public data sharing except upon request and approval by the Ministry of Health.

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
