# Peer review of "Implementation of an Effective Decentralised Programme for Detection, Treatment and Prevention of Tuberculosis in Children"

_tropicalmed, 2021, doi:10.3390/tropicalmed6030131_

Round 1

Reviewer 1 Report

This paper is very well written and represents a significant step towards prevention-treatment of tbc in children.

Author Response

Many thanks for your positive review.

Reviewer 2 Report

I have found the topic of the article very interesting. I agree with the authors that a rapid diagnosis of tuberculosis has to be carried out and for this, in addition to the clinical diagnosis, there is the microbiological diagnosis. Within the microbiological diagnosis, in addition to smear microscopy, there is GeneXpert MTB / RIF ultra, which, in addition to detecting Mycobacterium tuberculosis complex, detects resistance to rifampicin. Possibly if the GeneXpert MTB / RIF ultra had been used in this study, the diagnosis of tuberculosis would have been greater.

Author Response

Many thanks for your positive comments.

Uganda has been implementing Xpert (and now Ultra) for detection of MTB and Rif resistance for some years. At the time of this study, these tests were not available as a decentralised service. We agree that availability would increase detection and your point is added to discussion.

Reviewer 3 Report

This manuscript provided a details description of the DETECT Child TB project including its outcomes, challenges and lessons learnt. Most importantly, based on the outcome/lessons from this project it has already been implemented to other regions of Uganda and some other African countries.  This work will be useful to contribute significantly to reduce the burden of global TB. The manuscript is well-written and explained properly. Following minor issues need to be corrected before it gets accepted/published:

Page 1, line 38-Please write mycobacterium tuberculosis instead of TB because TB doesn’t cause disease. The diseases are caused by microorganisms such as mycobacterium tuberculosis.

Page 6, lines 123-127-Please correct font type and size and keep consistent throughout the manuscript.

Page 7, line 136-Please write the numbering of the headings/subheadings correctly. It should be 2 and correct all other numbering accordingly.

Page 8, figure 2-Please explain the abbreviation mentioned in the figure as a footnote or in the figure legend. For example, IPT, HH etc.

Page 9, line 189-Correct the numbering.

Author Response

Many thanks for your careful review.

All suggested edits have been addressed in the revised version of the manuscript.